# Pentavalent Disabled Infectious Single Animal (DISA)/DIVA Vaccine Provides Protection in Sheep and Cattle against Different Serotypes of Bluetongue Virus

**DOI:** 10.3390/vaccines9101150

**Published:** 2021-10-09

**Authors:** Piet A. van Rijn, Mieke A. Maris-Veldhuis, Massimo Spedicato, Giovanni Savini, René G. P. van Gennip

**Affiliations:** 1Department of Virology, Wageningen Bioveterinary Research (WBVR), 8200 RA Lelystad, The Netherlands; mieke.maris@wur.nl (M.A.M.-V.); rene.vangennip@wur.nl (R.G.P.v.G.); 2Department of Biochemistry, Centre for Human Metabolomics, North-West University, Potchefstroom 2520, South Africa; 3Public Health Department, Istituto Zooprofilattico Sperimentale dell’Abruzzo e del Molise “G. Caporale”, 64100 Teramo, Italy; m.spedicato@izs.it (M.S.); g.savini@izs.it (G.S.)

**Keywords:** bluetongue virus, vaccine platform, DISA, DIVA, NS3, serotype exchange

## Abstract

Bluetongue (BT) is a midge-borne OIE-notifiable disease of ruminants caused by the bluetongue virus (BTV). There are at least 29 BTV serotypes as determined by serum neutralization tests and genetic analyses of genome segment 2 encoding serotype immunodominant VP2 protein. Large parts of the world are endemic for multiple serotypes. The most effective control measure of BT is vaccination. Conventionally live-attenuated and inactivated BT vaccines are available but have their specific pros and cons and are not DIVA compatible. The prototype Disabled Infectious Single Animal (DISA)/DIVA vaccine based on knockout of NS3/NS3a protein of live-attenuated BTV, shortly named DISA8, fulfills all criteria for modern veterinary vaccines of sheep. Recently, DISA8 with an internal in-frame deletion of 72 amino acid codons in NS3/NS3a showed a similar ideal vaccine profile in cattle. Here, the DISA/DIVA vaccine platform was applied for other serotypes, and pentavalent DISA/DIVA vaccine for “European” serotypes 1, 2, 3, 4, 8 was studied in sheep and cattle. Protection was demonstrated for two serotypes, and neutralization Ab titers indicate protection against other included serotypes. The DISA/DIVA vaccine platform is flexible in use and generates monovalent and multivalent DISA vaccines to combat specific field situations with respect to Bluetongue.

## 1. Introduction

Bluetongue (BT) is a severe notifiable midge-borne disease of domestic and wild ruminants caused by the bluetongue virus (BTV) [1]. BT is characterized by fever, erosions, and cyanosis of the mucous membranes of the oronasal cavity, nasal discharge, edema of the head and neck, coronitis, laminitis, and anorexia [2,3]. Sheep species are the most susceptible among domestic livestock, while cattle are considered the main reservoir host for onward transmission of the virus to vectors. BT outbreaks lead to large economic losses related to diseased animals, production losses, and restrictions on trade and movements of ruminants from affected areas [4,5].

BTV is a distinct virus species within the genus *Orbivirus* of the family *Reoviridae* and is the prototype orbivirus [6]. The genus *Orbivirus* consists of at least 22 virus species of which midge-borne African horse sickness virus and enzootic hemorrhagic disease virus also cause notifiable disease of equids and ruminants, respectively [1]. The virion consists of a multilayered nonenveloped particle containing a 10-segmented genome of double-stranded RNAs (Seg-1 to -10) encoding seven structural proteins VP1-7 and at least four non-structural proteins NS1-4 [6,7,8,9]. NS3/NS3a protein encoded by Seg-10 is a multifunctional protein involved in virus release from the infected cell and immunosuppression of the ruminant host [10,11]. Seg-2 and Seg-6 codes for outer shell proteins VP2 and VP5, respectively, of which VP2 is the major serotype-specific protein inducing neutralizing antibodies (nAbs) [12].

The BTV species consists of typical and atypical serotypes that can be subdivided into so-called nucleotypes (Figure 1). The typical serotypes 1–24 are notifiable and are spread by bites of *Culicoides* midges [1,13,14]. Their occurrence is strictly associated with the presence of competent *Culicoides* species suggesting the concept of viral vector ecosystems [15,16,17,18]. Atypical BTV serotypes 25–27 have been exclusively found in small ruminants causing subclinical infection. For some of these atypical BTV strains, in-contact transmission has been demonstrated, while virus replication in culicoides cells has failed [19,20,21]. More recently, new serotypes are recognized and proposed [22,23,24,25,26,27,28].

Co-circulation of different serotypes has been observed in large parts of the tropical world and areas with temperate climates. BTV serotypes 1, 2, 3, 4, 6, 8, 9, 11, 14, and 16 have been reported in the European Union in the 21st century. Serotypes 1, 2, 3, 4, 8, and 16 are still circulating [29]. Vaccination is the most effective tool to control outbreaks. Marketed inactivated and conventionally live-attenuated vaccines (LAVs) are used in different parts of the world depending on the local epidemiological situation [30]. Both types of BT vaccines do not enable “Differentiating Infected from Vaccinated Animals” (DIVA principle) [31]. Mono- and bivalent inactivated BT vaccines for different serotypes are in use in Europe [32,33], but simultaneous control of many serotypes is complicated with inactivated vaccines. To achieve broad protection, three pentavalent LAV cocktails are in use in Africa. Cocktails have to be administered in the correct order due to the residual virulence of these conventionally attenuated BTV strains in the subsequent pentavalent LAV cocktails [34]. The safety of LAVs is controversial. These can cause clinical signs and uncontrolled spread by midges with the consequent possible occurrence of reversion to virulence and reassortment events between different LAVs or with wild-type BTV [35,36].

Next-generation BT vaccine candidates have been developed, which were reviewed in [37]. BTV-based vaccine platforms enable the exchange of serotype determining genome segments resulting in “serotyped” vaccine candidates [38,39,40,41]. Consequently, such BTV-based vaccines share eight or nine genome segments complemented with Seg-2 (VP2), or Seg-2 (VP2) and Seg-6 (VP5) of the respective serotypes. The Disabled Infectious Single Animal (DISA) vaccine platform is based on LAVs without functional NS3/NS3a protein, therewith blocking onward spread by midges by lack of viremia in ruminants and lack of propagation in competent midges (DISA principle) [42,43]. Importantly, the prototype DISA vaccine for serotype 8 induces complete early and serotype-specific lasting protection in sheep [44]. Furthermore, the DISA vaccine fulfills all criteria for modern veterinary vaccines [37], since it accomplishes safety with regard to disease, adverse effects, viremia in sheep, onward transmission by midges, and enables DIVA by an experimental NS3 cELISA and the OIE-recommended panBTV PCR test [45]. Recently, the DISA and DIVA principles have been demonstrated for an internal in-frame deletion of 72 amino acid codons (72-aa) in Seg-10 (NS3/NS3a) [46,47]. Similar to the prototype DISA vaccine for serotype 8 in sheep, this DISA/DIVA vaccine for serotype 8, shortly named DISA8, has shown an ideal vaccine profile in cattle.

Here, the DISA/DIVA vaccine platform was applied for different serotypes aiming for protection against multiple serotypes and eventually broad protection. As an example, pentavalent vaccine DISA12348, consisting of “European” serotypes 1, 2, 3, 4, and 8, was studied in sheep and cattle.

## 2. Materials and Methods

BSR cells (a clone of BHK-21 cells [48]) were cultured in Dulbecco’s modified Eagle’s medium (DMEM, Fischer Scientific, Landsmeer, The Netherlands), containing 5% fetal bovine serum (FBS), 100 IU/mL penicillin, 100 μg/mL streptomycin, and 2.5 µg/mL amphotericin B (Fischer Scientific, Landsmeer, The Netherlands).

BTV8/net07/e1/bhkp3 (BTV8) and BTV2/SAD01/01/e1/bhkp2/kcp3 (BTV2) were used as challenge viruses [44]. Standard reference BTV strains for different serotypes (obtained from Onderstepoort Veterinary Institute, Pretoria, South Africa) were used to determine serum neutralizing antibodies (nAbs) against the respective serotypes [49].

The BTV6/net08 strain [50,51] with an in-frame 72-aa deletion in Seg-10 (NS3/NS3a) and exchanged Seg-2 (VP2) and Seg-6 (VP5) originating from different serotypes were generated using reverse genetics as previously described [47]. Briefly, plasmids with optimized genes expressing BTV6 proteins in combination with the appropriate set of 10 run-off RNA transcripts were used [52]. cDNAs encoding run-off RNA transcripts of the DISA/DIVA vaccine platform have been described [47,53]. Similarly, cDNAs of Seg-2 (VP2) and Seg-6 (VP5) of different serotypes in an appropriate plasmid for run-off RNA synthesis were synthesized by Genscript Corporation Piscataway, NJ, USA (Table 1). In general, these cDNAs consist of open reading frames of the respective serotype and nontranslated regions originating from BTV6/net08. In addition, a previously described chimeric Seg-2 cDNA of serotypes 1 and 16 was used [41]. Attempts to rescue the virus were repeated at least twice in order to conclude that “serotype exchange” for the respective serotype was unsuccessful. For these serotypes, heterologous Seg-6 (VP5) was used to attempt the incorporation of Seg-2 (VP2) of the respective serotype (Table 1). The genome constellation of the rescued virus was confirmed (partial) sequencing of Seg-2, Seg-6, and the 72-aa deletion in Seg-10 according to standard procedures. DISA/DIVA vaccines for different serotypes were similarly named as prototype DISA8 according to the serotype of incorporated Seg-2 (VP2) (Table 1). DISA vaccine stocks were produced at low multiplicity of infection (MOI) of BSR cells. DISA vaccine was harvested by freeze thawing and centrifugation when >50% of cells were immunostained positive for VP7 in a duplicate well using anti-VP7 monoclonal antibody ATCC-CRL-1875. Virus titers were determined by endpoint dilution on BSR cells and expressed as 50% tissue culture infectious dose per mL (TCID_50_/mL). Vaccine stocks of clarified freeze/thaw lysates of DISA1, 2, 3, 4, and 8 were used to prepare pentavalent vaccine DISA12348.

Vaccination-challenge experiments were performed in sheep and cattle under the guidelines of the European Community and were approved by the Committee on the Ethics of Animal Experiments of Wageningen Bioveterinary Research (permit numbers 2017.D-0070.001 and 2017.D-0070.003). Animals were obtained from Dutch farms and were free of BTV and BTV antibodies. In total, 16 ewes (cross breed Noordhollander × Texel) of approximately 10 months old and 16 cows (cross breed Holstein Frisian × Belgian Blue) of about one year old arrived one week prior to the first vaccination or infection.

Equal amounts of DISA vaccines for serotypes 1, 2, 3, 4, and 8 were combined to formulate the pentavalent DISA12348 vaccine. For sheep, the total vaccine dose was 2 × 1 mL of 1 × 10^5^ TCID_50_/mL DISA12348 per vaccination and thus contained 0.2 × 10^5^ TCID_50_/mL DISA vaccine per serotype. Based on the similar seroconversion for VP7 Abs after immunization with a 10-fold or 100-fold lower vaccine, we expected protection with a 5-fold lower dose in pentavalent DISA12348. For cattle, the vaccine dose was intended to be five times higher consisting of 2 × 1 mL of 5 × 10^5^ TCID_50_/mL DISA12348 per vaccination, which corresponded to the standard dose for each serotype. Unfortunately, the titer of the vaccine stock of DISA3 was declined and too low to prepare DISA12348 with equal titers of 1 × 10^5^ TCID_50_/mL for each serotype. Consequently, DISA12348 used for prime vaccination contained 0.2 × 10^4^ TCID_50_/mL DISA3, which was 50 times less than for the other DISA vaccines. For boost vaccination, a newly prepared DISA3 stock was sufficiently high (5.8 × 10^5^ TCID_50_/mL) to formulate DISA12348 of 5 × 10^5^ TCID_50_/mL with equal titers of 1 × 10^5^ TCID_50_/mL DISA vaccine for each serotype.

The vaccination scheme for sheep and cattle was performed according to previous studies [44,47]. Briefly, two groups of four animals were intramuscularly (im) vaccinated left and right in the neck with 1 mL DISA12348 on day 0 and were similarly boost vaccinated on day 21 of the experiment. One vaccinated group and one naïve group, serving as a control group, were challenged on day 84, corresponding to day 0 post challenge (dpc), with virulent BTV2 or BTV8. BTV challenge was performed subcutaneously (sc) with 4 × 1 mL 1 × 10^5^ TCID_50_/mL BTV2 or BTV8 left and right in the neck for cattle, and in the back left and right from the spinal cord for sheep. The experiments were finalized at day 105, three weeks post challenge (21 dpc). Body temperature and clinical signs were monitored throughout the experiment. Clinical signs were scored according to the clinical score table for Bluetongue (Appendix A). Animals were blood sampled on indicated days of the experiment.

EDTA blood samples were examined for BTV RNA with panBTV PCR tests targeting Seg-1 and Seg-10 [47]. Crossing point (Cp) values were calculated, and samples without Cp value showing an increase in the OD_640/530_ were interpreted as 40 and negative samples were set at 45. It is worth noting that the Seg-10 panBTV PCR test specifically detects wild-type BTVs and serves as accompanying the DIVA PCR test of DISA vaccines.

Serum samples were examined for VP7 antibodies (Abs) with a Bluetongue VP7 competition enzyme-linked immunosorbent assay (VP7 cELISA) (ID.Vet, Grabels, France), and the percentage of blocking was displayed as 100 minus value. The threshold was 50% blocking according to the supplier. NS3-directed Abs were detected with the experimental NS3 cELISA [54]. The percentage of blocking was displayed as 100 minus value and an arbitrary threshold of 50% was used [47]. It is worth noting that the NS3 cELISA specifically detects BTV-infected animals and serves as accompanying DIVA ELISA of DISA vaccines.

Sera on days 84 and 105 of the experiment corresponding to 0 and 21 dpc were examined for neutralizing Abs (nAbs) against serotypes 1, 2, 3, 4, and 8 with the serum neutralization test (SNT) [49,55]. Briefly, sera were 30 min heat-inactivated at 56 °C, and then 2-fold serial dilutions of 50 µL from 2 to 256 were made in minimal essential medium (MEM, Sigma, Milan, Italy) with antibiotics and FBS. Diluted sera were incubated with an equal volume of 100–300 TCID_50_ of the OIE standard reference BTV strain of the appropriate serotype. After one hour at 37 °C in 5% CO_2_, approximately 10^4^ Vero cells in 50 µL of MEM were added per well and incubation was continued. Starting after three days, wells were scored for cytopathic effect (CPE). The nAb titer was expressed as the highest dilution showing more than 75% CPE neutralization in the Vero cell monolayer.

## 3. Results

### 3.1. Rescue of DISA/DIVA Vaccines

DISA/DIVA vaccines, here shortly named DISA vaccines, for different serotypes were rescued using reverse genetics. All DISA vaccines contain the backbone of LAV strain BTV6/net08 with the in-frame 72-aa deletion in Seg-10 (NS3/NS3a) [47]. Thus, DISA vaccines share 8 out of 10 genome segments. The genome constellation was completed with Seg-2 (VP2) and Seg-6 (VP5) of different serotypes. Accordingly, DISA vaccines for serotypes 1, 2, 3, 4, 6, 8, 17, and 25 were rescued (Table 1). DISA2 with Seg-2 (VP2) and Seg-6 (VP5) of European/African BTV2 isolates was rescued, but the virus titer rapidly declined after virus passages (not shown). Alternatively, DISA2 with Seg-2 (VP2) and Seg-6 (VP5) of BTV2 from the USA was rescued and grew to a normal virus titer (Table 1). DISA3 showed a lower virus titer of 1 × 10^5^ TCID_50_/mL and seemed to be unstable in time (not shown). Attempts to rescue DISA vaccines for serotypes 7, 9, 14, 16, 22, and 29 failed. By a second approach, representatives of DISA9 and DISA14 with their serotype immunodominant Seg-2 (VP2) were rescued in combination with heterologous Seg-6 (VP5) (Table 1). Unfortunately, this approach also failed for the other tested serotypes. A potential representative of DISA16 was rescued by the use of chimeric Seg-2 (VP2) (Table 1).

Genome constellations of vaccine stocks were confirmed by partial sequencing of exchanged genome segments 2 and 6. The presence of the 72-aa deletion was confirmed by sequencing of the entire open reading frame of Seg-10. As expected, DISA vaccines were detected with the Seg-1 panBTV PCR test and not with the Seg-10 panBTV PCR test (DIVA PCR test) [47] (not shown). Typically, prepared vaccine stocks had titers of 0.8–1.4 × 10^7^ TCID_50_/mL, whereas DISA3 had an approximately 100-fold lower virus titer of 1 × 10^5^ TCID_50_/mL. An extra passage on BSR cells resulted in a higher virus titer of 5.8 × 10^5^ TCID_50_/mL of the freshly prepared virus stock, which is still >10 times lower than for the other DISA vaccine stocks.

### 3.2. Animal Trials

#### 3.2.1. Vaccination-Challenge Experiment in Sheep

No clinical signs, fever, local reactions, or other abnormalities related to vaccination were observed following vaccination (not shown). Further, vaccinated groups did not show clinical signs after challenge with virulent BTV2 or BTV8, except for one sheep in both groups showing loss of appetite (score of 1) for several days in the third week after challenge (Figure 2). Remarkably, vaccinated sheep 2666 challenged with BTV2 showed fever (41.1 °C) and loss of appetite on 5 dpc. In contrast, control sheep developed clinical signs such as fever (>40 °C), depression, increased breath frequency, and loss of appetite between 5 to 21 dpc (Figure 2).

DISA12348 vaccinations resulted in extremely high average Cp values of >40 with the Seg-1 panBTV PCR test (Figure 3). PCR results for individual animals reached Cp values of ±37, and boost vaccination did not lead to lower Cp values (not shown). It is worth noting that DISA vaccines cannot be detected with the Seg-10 panBTV PCR test (DIVA PCR test). Therefore, samples prior to the BTV challenge were not tested with the DIVA PCR test. After the BTV challenge, PCR positivity was detected in control groups with the DIVA PCR test and the panBTV Seg-1 PCR test (Figure 3). Except for one BTV8-infected control sheep, which remained PCR negative (not shown), all animals of both control groups were PCR positive from 3 to 21 dpc. BTV challenge of vaccinated groups did not result in viremia as detected by the DIVA PCR test (Figure 3), except for sheep 2666, which developed Cp values of ±32 between 5 and 7 dpc. Apparently, sheep 2666 was not completely protected and solely contributed to the mean PCR positivity of this vaccinated group after the BTV2 challenge.

BTV challenge was also monitored with the Seg-1 panBTV PCR test detecting wild-type BTVs and DISA vaccines (Figure 3). The vaccinated group was PCR negative (Cp = 45) from day 49 to day 84 but developed PCR positivity after the BTV2 challenge. PCR positivity was mainly but not exclusively caused by sheep 2666, since several sheep showed Cp values of 40 for one or more days after the BTV2 challenge. The vaccinated group challenged with BTV8 was not completely PCR negative on the day of the BTV8 challenge. The high Cp values post challenge were likely remains of DISA12348 vaccine since no decrease of the PCR signal was observed after the BTV8 challenge, suggesting that virulent BTV8 did not cause viremia. To summarize, results of the Seg-1 panBTV PCR test confirmed these with the DIVA PCR test.

Both vaccinated groups seroconverted to >50% blocking for VP7 Abs in the third week after prime vaccination (Figure 4). It is worth noting that one vaccinated group showed ±60% blocking from 10 to 17 dpc and further increased to 87% at 21 dpv, the day of boost vaccination. A small increase to the maximum of 95–100% blocking was observed for this group after boost vaccination, which remained high up to and after the BTV8 challenge. Remarkably, seroconversion of the other group of four sheep remained 65–70% from 10 to 21 dpc and did not further increase after boost vaccination but reached rapidly the maximum of 95–100% after the BTV2 challenge (Figure 4). Sheep 2667 and 2669 of this vaccinated group showed a lower seroconversion of 40–50% at 84 dpv, compared with the other six vaccinated sheep (not shown). Both sheep rapidly seroconverted to ±100% blocking after the BTV2 challenge. For vaccinated sheep showing +100% blocking on 0 dpc, no increase of seroconversion could be measured after the challenge. Finally, challenge control groups rapidly seroconverted in the second week after the BTV challenge to ±100% blocking in the VP7 cELISA, which was faster than after DISA vaccination (Figure 4).

Control groups also seroconverted for NS3 Abs in the second week after the BTV challenge, whereas DISA12348-vaccinated groups remained seronegative (<50% blocking) up to day 84, the day of challenge (0 dpc) (Figure 4). BTV2 challenge of vaccinated sheep resulted in rapid NS3 seroconversion at 7 dpc and remained 55–65% (seropositive) up to 21 dpc. Sheep 2666 significantly contributed to the mean % blocking for NS3 Abs (>70% at 11–19 dpc); however, two other vaccinated sheep of this group were also seropositive (>50%) starting at 7 dpc for several days (not shown). The vaccinated group challenged with BTV8 showed a very small increase (Figure 4), but all four vaccinated sheep remained negative (<50% blocking) for NS3 Abs in the experimental NS3 cELISA.

#### 3.2.2. Vaccination-Challenge Experiment in Cattle

No elevated body temperature, clinical signs, local reactions, or other abnormalities were observed after vaccination. Conversely, naïve cattle challenged with BTV8 showed mild clinical signs, such as loss of appetite and salivation for one or two days between 3 and 8 dpc (not shown). Prime vaccination resulted in very high average Cp values (>39) with the Seg-1 panBTV PCR test (Figure 5). PCR positivity after vaccination was measured on day 7 for the first time, and all cattle were PCR positive on day 21, the day of boost vaccination. Boost vaccination did not increase the amount of viral RNA, as determined by PCR testing. Even more, the average PCR signal increased slowly after day 28 and steadily more cattle became PCR negative (Cp = 45). Still, one animal showed Cp = 40 on day 84, the day of the challenge. More importantly, the lowest Cp value detected in vaccinated animals was 36. EDTA blood samples before the BTV challenge were not tested with the Seg-10 panBTV PCR test, since DISA12348 is not detected by this accompanying DIVA PCR test.

After the BTV challenge, cattle were monitored with the DIVA PCR test (Seg-10 panBTV PCR test) (Figure 5). As expected, control groups developed mean Cp values up to 28–30 from 3–4 dpc to the end of the experiment (21 dpc), which confirmed successful infection with virulent BTV2 and 8. One control animal infected with BTV8 developed doubtful PCR results (Cp = 40) between 10 and 12 dpc. This very low PCR positivity was confirmed by the Seg-1 panBTV PCR test, showing Cp = 40 on 6–10 dpc (not shown). Vaccinated groups remained completely negative with the DIVA PCR test after the BTV2 or BTV8 challenge (Figure 5). Results with the Seg-1 panBTV PCR test were negative or Cp = 40, confirming that both challenge viruses did not replicate in DISA12348 vaccinated groups (Figure 5). Apparently, DISA12348-vaccinated cattle showed sterile immunity against serotypes 2 and 8.

Cattle were monitored for Abs directed against VP7 throughout the experiment (Figure 6). Control groups rapidly seroconverted (>50%) in the second week and reached the maximum of ±100% blocking in the third week after the BTV challenge. Both vaccinated groups seroconverted slower and became positive (>50%) for VP7 Abs in the third week after prime vaccination. VP7 seroconversion of the vaccinated groups was slightly different due to rapid seroconversion of two animals in the vaccinated group challenged with BTV2 later on. After boost vaccination, and after challenge, no further increase in blocking percentage could be measured with the VP7 cELISA.

DISA12348-vaccinated groups were still negative for NS3 Abs on 84 dpv/0 dpc (blocking of <50%), although both vaccinated groups showed a slightly higher blocking of 30–40% at 84 dpv than control cattle (20%) at 0 dpc (Figure 6). One animal showed 60% blocking on day 84, which was interpreted as seropositive. BTV2 or BTV8 challenge of vaccinated cattle did not increase the percentage of blocking for NS3 Abs. In contrast, control groups rapidly seroconverted in the second week after the BTV challenge. NS3 seroconversion of the BTV2 control group was slower and weaker than of the BTV8 control group.

### 3.3. Neutralizing Abs by Pentavalent DISA12348 Vaccination

Serum neutralization test (SNT) was performed for sera collected on day 84 (0 dpc) and day 105 (21 dpc) of sheep and cattle (Figure 7). Neutralizing Abs (nAbs) were determined for all five serotypes included in the pentavalent DISA12348 vaccine. Control sheep were negative for nAbs on 0 dpc (day 84) and developed nAbs against serotype 2 (≥256) and serotype 8 (4–32) on 21 dpc, day 105 (Figure 7, left panel). nAb titers were highly serotype specific since cross-neutralizing Abs were not measured, except for two sheep showing serotype 1 specific nAb titers of 8 after the BTV2 challenge.

nAb titers on 84 dpv of DISA12348 vaccinated sheep varied largely between sheep as well as for different serotypes (Figure 7, left panel). Seven out of eight vaccinated sheep had serotype 1 specific nAb titers of 8–64 on 84 dpv, but nAb titers for other serotypes were very low (0–16). BTV2 challenge of vaccinated sheep increased serotype 2 specific nAb titers from one sheep with a nAb titer of 2 on day 84 to all four sheep with nAbs of 32 to >256 on day 105, three weeks post challenge. BTV8 challenge resulted in serotype 8 specific nAb titers of 2–16 at day 105 (21 dpc), while nAbs were marginally detected on day 84. nAb titers against other serotypes remained low and did not increase after BTV2 or BTV8 challenge. Remarkably, nAb titers of the vaccinated-challenged sheep were slightly lower on day 105 (21 dpc), compared with nAb titers of naïve-challenged sheep (Figure 7, left panel).

Control cattle were negative for nAbs against serotypes 1 and 8 on 0 dpc, day 84 of the experiment (Figure 7, right panel). Surprisingly, control cattle were not completely negative for nAbs against serotypes 2, 3, and 4. We assumed that these measured nAb titers on 0 dpc were not specific, since cattle were free of BTV-VP7 Abs on day 84 (0 dpc) (Figure 6). After the BTV challenge, control cattle developed nAb titers of 128 to >256, and 8–256 on 21 dpc (day 105) against serotypes 2 and 8, respectively. Very low nonspecific nAbs titers against serotypes 1, 3, and 4 were detected on day 105.

After prime-boost DISA12348 vaccination, nAb titers on day 84 varied between different serotypes from 0–8 against serotypes 2 and 3 to 8–256 against serotypes 1, 4, and 8 (Figure 7, right panel). BTV2 challenge of vaccinated cattle resulted in increased serotype 2 specific nAb titers from 2–8 on day 84 to 8–64 on day 105 (21 dpc). Similarly, the BTV8 challenge of vaccinated cattle resulted in increased serotype 8 specific nAb titers from 8–32 on day 84 to 64–128 on day 105. nAb titers against serotypes 1, 3, and 4 did not increase after BTV2 or BTV8 challenge. Serotype 2 specific nAb titers on day 105 (8–64) in vaccinated-challenged cattle were lower than in naïve BTV2-challenged cattle (128 to >256) on day 105 (21 dpc) (Figure 7, right panel).

## 4. Discussion

The DISA/DIVA vaccine platform was explored for several serotypes by exchange of Seg-2 (VP2), or Seg-2 (VP2) and Seg-6 (VP5). DISA vaccines for serotypes 1, 2, 3, 4, 6, 8, 17, and 25 were generated (Table 1). Our results are in agreement with previously published results being successful for 16 out 26 studied serotypes [40]. The exchange of serotype-specific outer shell proteins is flexible but not applicable for all BTV serotypes. A broader application was achieved by incorporation of Seg-2 (VP2) in combination with heterologous Seg-6 (VP5) and by incorporation of chimeric Seg-2 (VP2) (Table 1) [41]. The success of serotype exchange does not depend on the serotype per se, as shown for serotype 2 (Table 1). Further rescued DISA vaccine can show different characteristics such as a lower virus titer [40]. Detailed knowledge of interactions of the core particle with outer shell proteins, and their role in the infection process will facilitate the development of suitable DISA vaccines for more serotypes.

DISA vaccines for different serotypes are equally safe, as these share the vaccine platform consisting of eight genome segments of live-attenuated BTV6/net08 [53], including the 72-aa deletion Seg-10 (NS3/NS3a) [47]. Consequently, cocktails of DISA vaccines are completely safe, as has been demonstrated for DISA8 [47]. Further, reassortants consisting of genome segments of DISA vaccine and wtBTV will be less virulent than ancestor wtBTV. More importantly, the DISA vaccine replicates locally in ruminants and does not induce viremia (this study) [42,44,47], and cannot propagate in competent midges [43,46]. Taken together, the DISA vaccine is completely safe with respect to reassortment events since the chance on doubly infected cells, as required for reassortment between the DISA vaccine and wtBTV, is negligible.

Complete protection has been demonstrated by vaccination with the standard vaccine dose of DISA8 in sheep and cattle [42,44,47]. Here, pentavalent DISA12348 also induced complete protection in cattle for studied serotypes 2 and 8. Notably, DISA2 contained the outer shell proteins of BTV serotype 2 from the USA, while protection was studied with virulent BTV2 from Sardinia, Italy. nAb titers in cattle on day 84/0 dpc were similarly high for serotypes 1, 4, and 8 (Figure 7), suggesting protection against serotypes 1 and 4. The nAb titer for serotype 3 was obviously lower and did not justify expected protection. The intended dose of DISA3 used for prime vaccination of cattle was 50 times lower than for the other included serotypes, and DISA3 seemed to be unstable in vitro. We suggest that virus growth in vitro reflects the potency of DISA vaccine candidates in vivo.

In sheep, one-fifth of the standard vaccine dose per serotype was studied to gain insight with regard to the protective vaccine dose. nAbs were hardly detected on day 84, except for serotype 1 (Figure 7). Still, DISA12348 vaccination completely protected sheep against BTV8, and three out of four sheep were protected against BTV2 (Figure 2). Generally, VP7 Abs and nAbs indicate the level of the humoral response and the protective capacity of nAbs. Apparently, low or no detectable nAb titer does not mean “no protection” per se. Possibly, cell-mediated immunity plays an important role in protection after DISA vaccination. In contrast to serotype 8, protection against serotype 2 was not complete. The high nAb titer for serotype 1 assumed protection against virulent BTV1, whereas protection against serotypes 3 and 4 is not likely because of the very low nAb titers. We conclude that one-fifth of the standard vaccine dose per serotype in the multivalent DISA vaccine is not sufficient for complete protection, in contrast to the standard vaccine dose of 2 mL of 1 × 10^5^ TCID_50_/mL for each serotype. This standard vaccine dose is cost competitive, compared with marketed LAVs [34], and 100 times lower than of the Disabled Infectious Single Cell/Cycle (DISC) vaccine candidate [39,56] and the amount of antigen in inactivated BT vaccines [57].

VP7 seroconversion in sheep and cattle was slower after DISA12348 vaccination than after the BTV challenge. Remarkably, the increase of VP7 Abs slowed down from 10 to 17 dpv in both sheep and cattle and then further increased in the third week to the maximum blocking at 21 dpv. This was in agreement with previous studies after monovalent DISA8 vaccination of cattle [47]. Likely, the IgM to IgG switch after DISA vaccination differs from that after the BTV challenge [58]. NS3/NS3a protein inhibits the IFN response through several pathways, reviewed in [10]. We suggest that immunosuppression has switched off, which affects the induction of antibodies. Further, seroconversion remained high up to the day of challenge (day 84) after im boost vaccination, in contrast to prime-boost vaccination by the sc route [44]. More research is needed to elucidate the underlying mechanisms of the immune responses after DISA vaccination.

Previously, differentiation of infected from DISA vaccinated animals (DIVA principle) has been demonstrated by the Seg-10 panBTV PCR test and the DIVA NS3 cELISA [44,47,54,59]. Therefore, DISA12348 vaccination was not monitored with accompanying DIVA tests, and results were indeed negative on day 84 after prime-boost vaccination. Challenging control animals with BTV2 or BTV8 challenge resulted in PCR positivity and seroconversion by both DIVA tests, which clearly confirmed the DIVA principle. As previously discussed [47], the experimental NS3 cELISA requires improvement, in particular, because of partial NS3 seroconversion in cattle. Apparently, despite the 72-aa deletion encompassing the epitope of the competing MAb used in the NS3 cELISA [54], DISA vaccination led to interference for binding in the NS3 cELISA.

After the BTV challenge, DISA12348 vaccinated cattle did not seroconvert for NS3 Abs and did not show viremia by the DIVA PCR test (Figure 5 and Figure 6). As previously shown for DISA8 [47], DISA12348 vaccinated cattle here showed serotype-specific sterile immunity for studied serotypes 2 and 8 (Figure 7). Sterile immunity in sheep was also shown for serotype 8, whereas several sheep were not completely protected against BTV2. Indeed, incomplete protection resulted in seroconversion for NS3 Abs (Figure 3 and Figure 4). Sterile immunity was also confirmed by the highly sensitive OIE-recommended panBTV PCR test targeting the 72-aa region, which is deleted in Seg-10 of DISA/DIVA vaccines. Sterile immunity, as shown by negative results with both DIVA tests, is a strong indicator of the highest level of protection.

## 5. Conclusions

The DISA/DIVA vaccine platform shows improvements with respect to marketed vaccines [37]. Previously, the pros and cons of the platform, compared with other experimental BT vaccines for different field situations, have been extensively discussed [30]. More recently, the vaccine profile, including the DISA and DIVA principles, based on the in-frame 72-aa deletion in Seg-10 (NS3/NS3a) has been confirmed [46,47]. Here, the DISA/DIVA vaccine platform was applied for different serotypes. Pentavalent DISA12348 vaccine showed sterile immunity in sheep and cattle for studied serotypes 2 and 8 by lack of viremia and seroconversion against the DIVA target. Likely, protection was achieved for serotypes 1 and 4 based on nAb titers. Detailed knowledge of core–out shell interactions will be required to develop DISA vaccines for eventually all BTV serotypes. DISA vaccines will be suitable to combat specific emerging serotypes, to formulate tailormade multivalent DISA vaccines, and eventually, to formulate one or more multivalent DISA vaccines to achieve protection against all BTV serotypes.

## Figures and Tables

**Figure 1 vaccines-09-01150-f001:**
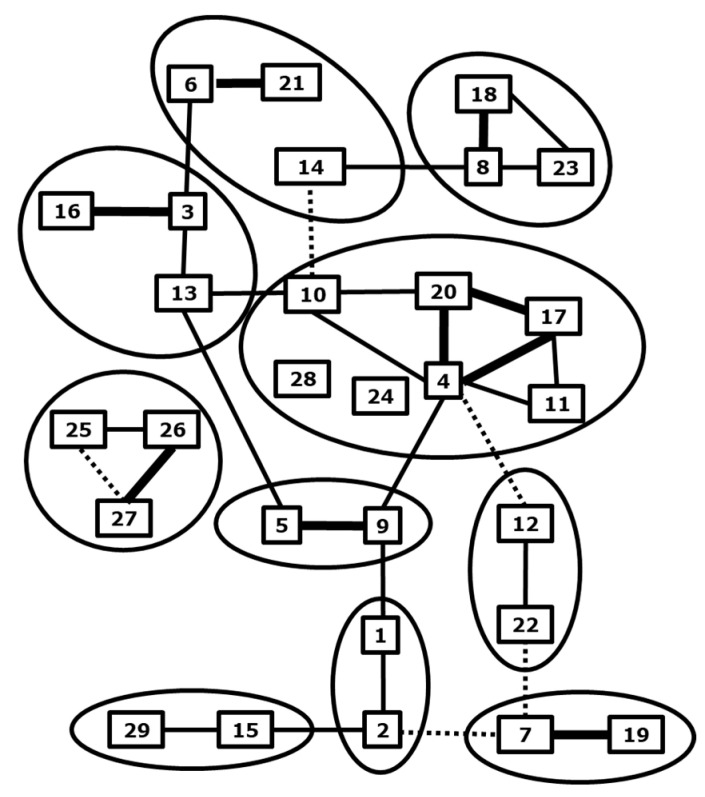
BTV serotypes. Cross-neutralization between BTV serotypes is arbitrarily quantitated by lines; strong (thick), some (normal) and weak (dashed). Serotypes are subdivided into “nucleotypes” by cross-neutralization assays (lines) and sequence analysis of Seg-2 (VP2) (circles) [13]. Updated for BTV27-29 [22,25].

**Figure 2 vaccines-09-01150-f002:**
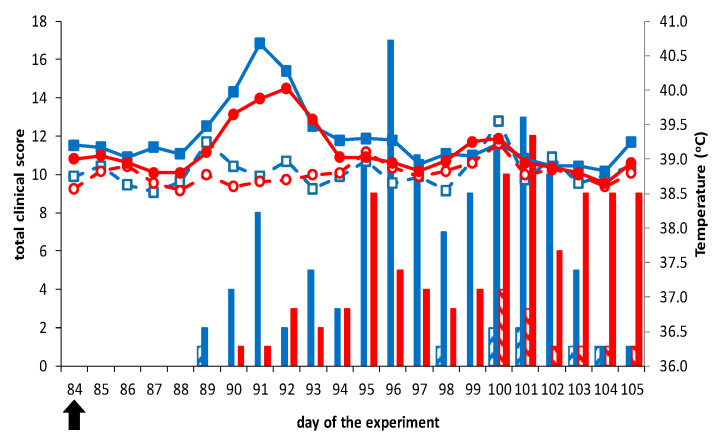
Body temperature and clinical signs in sheep. Prime-boost DISA12348 vaccinated groups and two challenge control groups were challenged on day 84 of the experiment (0 dpc) (arrow) with virulent BTV2 (blue, squares) or virulent BTV8 (red, circles). The average daily body temperature per group was calculated for vaccinated groups (open symbols) and control groups (filled symbols). Clinical signs were quantitatively monitored and the total clinical score for each group per day was calculated for vaccinated groups (striped bars) and control groups (filled bars). Neither elevated body temperature nor clinical signs were observed after the DISA12348 vaccination (not shown).

**Figure 3 vaccines-09-01150-f003:**
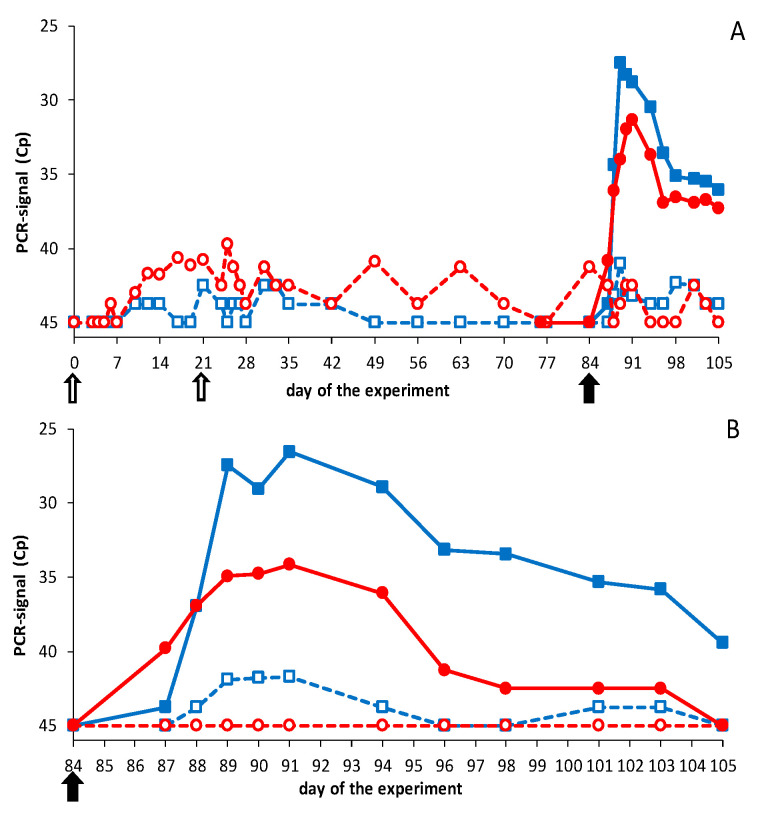
Results of the Seg-1 and Seg-10 panBTV PCR test in sheep. Groups were DISA12348 vaccinated three weeks apart on days 0 and 21 (white arrows). Vaccinated groups (open symbols) and control groups (filled symbols) were challenged on day 84 of the experiment (black arrow) with BTV2 (blue squares) or BTV8 (red circles): (**A**) average Cp value per group for the Seg-1 panBTV PCR test; (**B**) average Cp value per group for the DIVA PCR test (Seg-10 panBTV PCR test). PCR-negative results were set at 45. Vaccinated groups were negative for the DIVA PCR test before the BTV challenge (not shown).

**Figure 4 vaccines-09-01150-f004:**
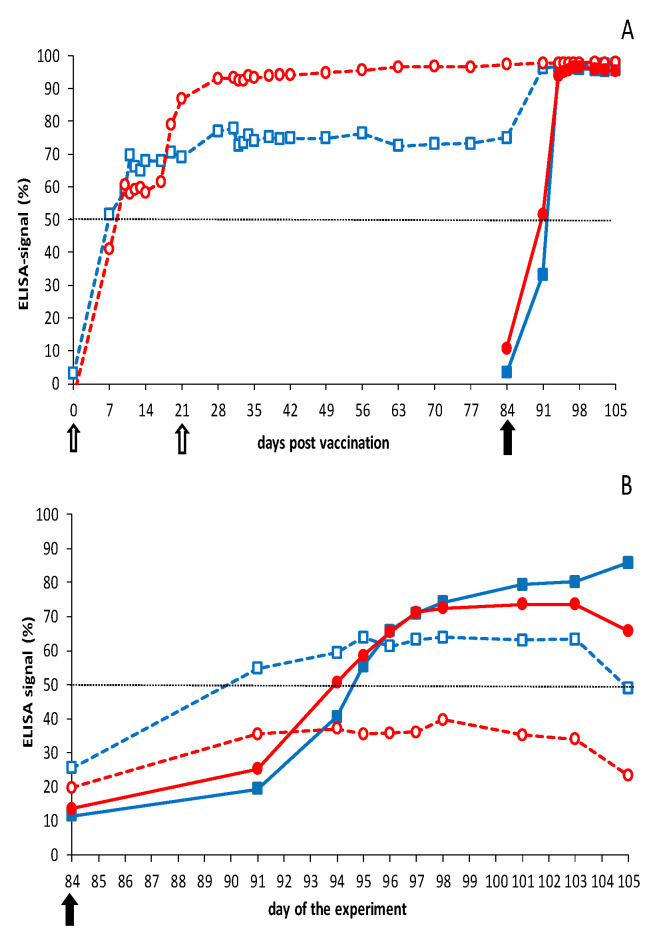
Results of the VP7 and NS3 cELISA in sheep. Groups were DISA12348 vaccinated twice three weeks apart on days 0 and 21 (white arrows). Vaccinated groups (open symbols) and control groups (filled symbols) were challenged on day 84 of the experiment (black arrow) with BTV2 (blue squares) or BTV8 (red circles): (**A**) average blocking % per group for the VP7 cELISA; (**B**) average blocking % per group for the DIVA ELISA (NS3 cELISA). Vaccinated groups were negative for the DIVA ELISA before the BTV challenge (not shown).

**Figure 5 vaccines-09-01150-f005:**
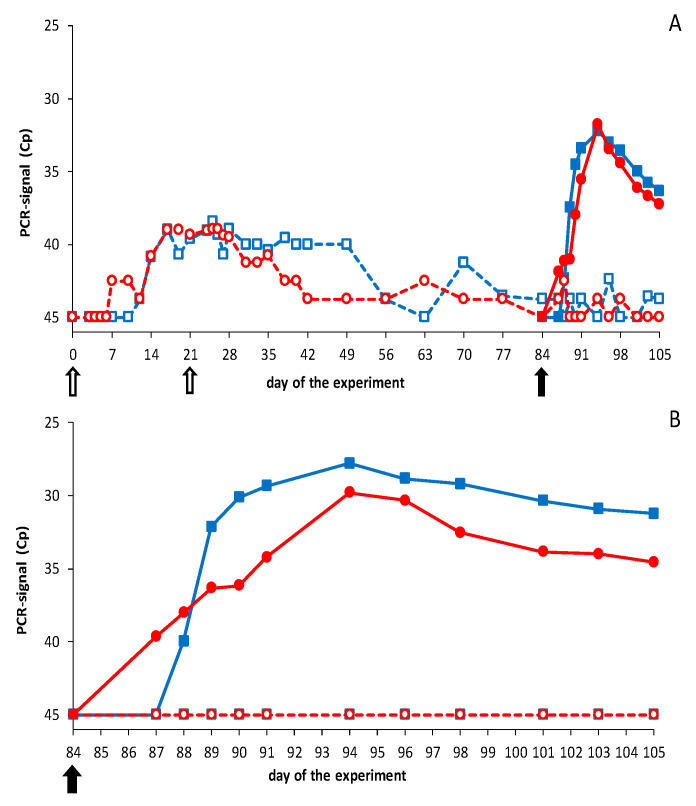
Results of the Seg-1 and Seg-10 panBTV PCR test in cattle. Groups were DISA12348 vaccinated twice three weeks apart on days 0 and 21 (white arrows). Vaccinated groups (open symbols) and control groups (filled symbols) were challenged on day 84 of the experiment (black arrow) with BTV2 (blue squares) or BTV8 (red circles): (**A**) average Cp value per group for the Seg-1 panBTV PCR test; (**B**) average Cp value per group for the DIVA PCR test (Seg-10 panBTV PCR test). PCR-negative results were set at 45. Vaccinated groups were negative for the DIVA PCR test before the BTV challenge (not shown).

**Figure 6 vaccines-09-01150-f006:**
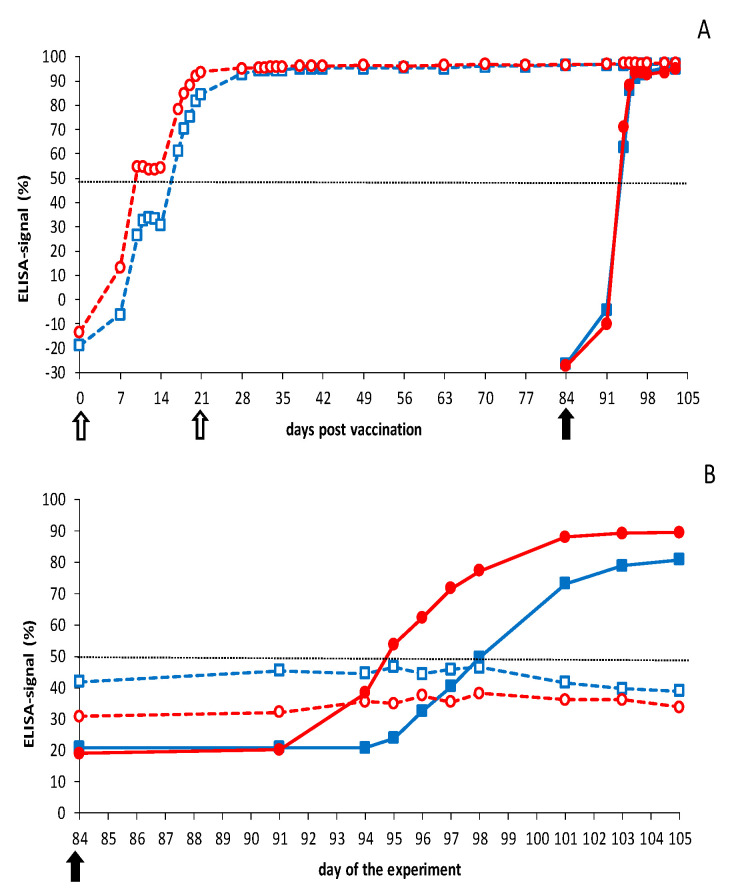
Results of the VP7 and NS3 cELISA in cattle. Groups were DISA12348 vaccinated twice three weeks apart on days 0 and 21 (white arrows). Vaccinated groups (open symbols) and control groups (filled symbols) were challenged on day 84 of the experiment (black arrow) with BTV2 (blue squares) or BTV8 (red circles): (**A**) average blocking % per group for the VP7 cELISA; (**B**) average blocking % per group for the DIVA ELISA (NS3 cELISA). Vaccinated groups were negative for the DIVA ELISA before the BTV challenge (not shown).

**Figure 7 vaccines-09-01150-f007:**
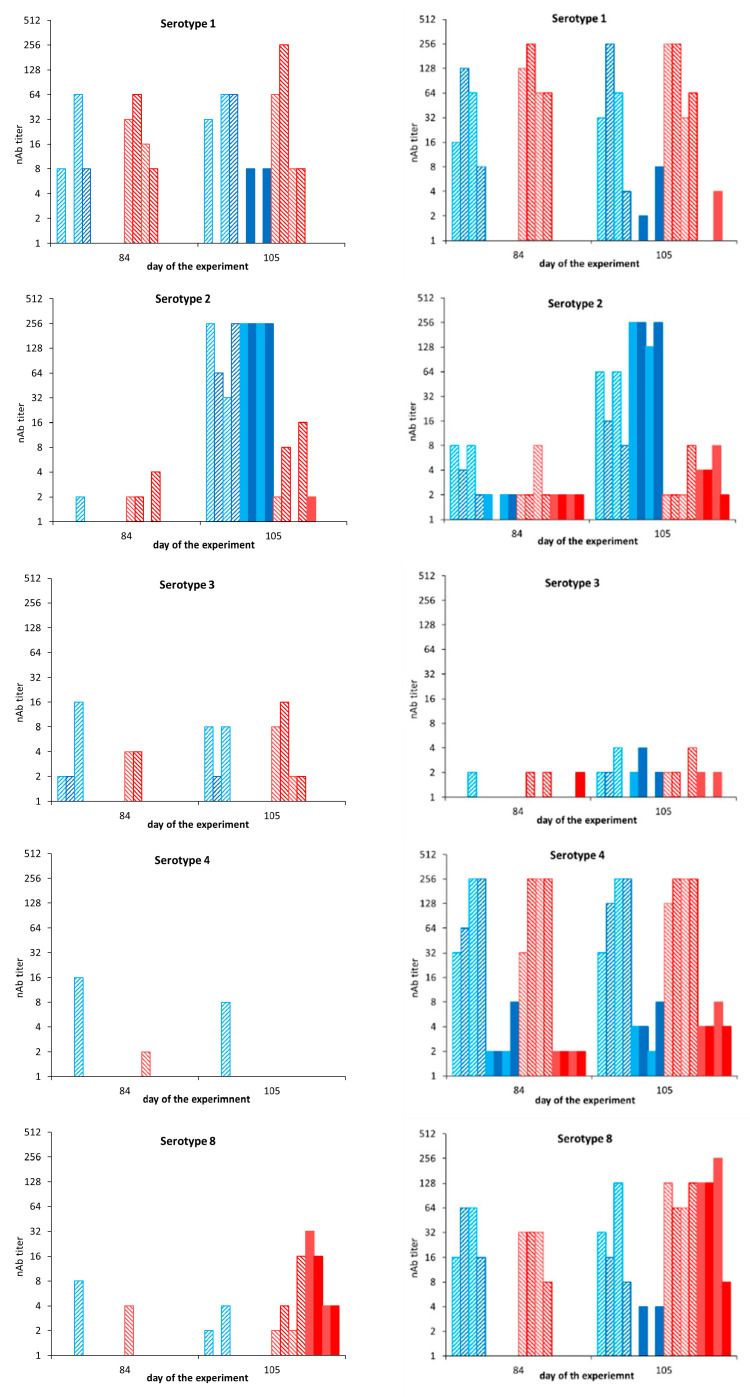
nAb titers by SNTs against indicated serotypes for sheep sera (**left panel**) and cattle sera (**right panel**) of days 84 and 105 of the experiment. Vaccinated groups (striped bars) and control groups (filled bars) were challenged with virulent BTV2 (blue) or virulent BTV8 (red) on day 84 of the experiment. Sera of days 84 (0 dpc) and 105 (21 dpc) were tested for nAbs against serotypes 1, 2, 3, 4, and 8. The highest serum dilution preventing 75% CPE formation in Vero cells is indicated. A prime-boost vaccination of sheep was performed with one-fifth of the standard vaccine dose, and the DISA12348 vaccine used for prime vaccination of cattle contained a lower dose of DISA3.

**Table 1 vaccines-09-01150-t001:** DISA/DIVA vaccines for different serotypes. DISA vaccines are named according to the serotype of Seg-2 (VP2). Exchanged genome segments 2 and 6 are indicated by serotype, origin (superscript), and NCBI accession numbers. Virus rescue is indicated by (+) or (−). The titer of freshly prepared virus stocks of DISA1, 2, 3, 4, and 8 are expressed in ^10^log TCID_50_/mL Note; DISA2 with Seg-2 (VP2)^IT^ and Seg-6 (VP5)^RSA^ was rescued but was unstable (+).

DISA Vaccine	Outer Shell		NCBI Accession nr.		
	VP2	VP5	Seg-2[VP2]	Seg-6[VP5]	Rescue
1	1	1	FJ969720	FJ969723	7.3
2	2^IT^	2^RSA^	JN255863	AJ586696	+
2	2^USA^	2^USA^	JQ822249	JQ822253	7.1
3	3^RSA^	3^RSA^	MG255540	AJ586697	5.8
4	4	4	AJ585125	AJ586699	6.8
6	6	6	FJ183375	FJ183379	+
7	7	6	AJ585128	FJ183379	-
8	8	8	GQ506452	GQ506456	6.9
9	9	9	AJ585130	AJ586708	-
9	9	1	AJ585130	FJ969723	+
14	14	6	AJ585135	FJ183379	+
16	16	16	FJ969720	FJ969723	-
16	16	6	AJ585137	FJ183379	-
16	16	1	FJ969720	FJ969723	-
1/16	1(16)	1	FJ969720/AJ585137	FJ969723	+
17	17	17	AJ585138	AJ586720	+
22	22	6	AJ585143	FJ183379	-
25	25	25	EU839840	EU839842	+
29	29	29	KP196604	KP196608	-
29	29	6	KP196604	FJ183379	-

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
