# Peer review of "Pentavalent Disabled Infectious Single Animal (DISA)/DIVA Vaccine Provides Protection in Sheep and Cattle against Different Serotypes of Bluetongue Virus"

_vaccines, 2021, doi:10.3390/vaccines9101150_

Round 1

Reviewer 1 Report

This is an interesting article reporting the investigations on pentavalent Disabled Infectious Single Animal (DISA)/DIVA vaccine intended to provide protection in sheep and cattle against different serotypes of bluetongue virus.

Live-attenuated and inactivated BT vaccines are conventionally available but have their specific pros and cons and are not DIVA compatible. The prototype Disabled Infectious Single Animal (DISA)/DIVA vaccine based on knockout of NS3/NS3a protein of live-attenuated BTV, named DISA8, fulfils all criteria for modern veterinary vaccines of sheep.

Recently, DISA8 with an internal in-frame deletion of 72 amino acid codons in NS3/NS3a showed a similar ideal vaccine profile in cattle.

In this study, the DISA/DIVA vaccine platform was applied for other serotypes, and pentavalent DISA/DIVA vaccine for “European” serotypes 1, 2, 3, 4, 8 was investigated in sheep and cattle.

Protection was demonstrated for two serotypes, and neutralization Ab titres indicate protection against other included serotypes.

It appears that the DISA/DIVA vaccine platform is flexible in use and can generate monovalent and polyvalent DISA vaccines to fight specific field situations with respect to Bluetongue.

The introduction is adequate.

The methodology is properly described.

The results obtained are well presented and clear despite the complexity of the topic.

Figures help to explain and understand the development and the evaluation process of the investigation.

Future applications of the acquired knowledge are presented.

Author Response

We like to thank reviewer 1 for the compliments about our manuscript.

Further, we checked the manuscript to improve the language and remove type errors as also requested by the other reviewers.

Reviewer 2 Report

Bluetongue virus (BTV) is the causative agent of an important livestock disease. To date, up to 24 classical serotypes of BTV have been described, and its distribution is expanding globally. Classical vaccine approaches, including live-attenuated and inactivated vaccines, have been used as prophylactic measures to control BTV infections. However, these vaccine approaches fail to address important matters like induction of a cross-protective immune response among different serotypes circulating in the same area. Moreover, the implementation of a DIVA strategy has become in an important issue for vaccines against pathogens relevant in animal health. In this manuscript, authors have described a pentavalent vaccine for serotypes1, 2, 3, 4 and 8, which are circulating in Europe. The vaccine is based in a previously described approach (DISA) based on knockout of NS3/NS3a protein, which was effective against one single serotype. In addition, this vaccine prototype could be DIVA compatible, although this will required additional studies. The manuscript is well written and describes an interesting approach for the development of multiserotype vaccines against BTV. However, certain points must be addressed and clarified:

  • Authors have used different vaccine dose for Sheep and cattle. The reason for that should be indicated in the text, eg: dose based in previous studies with the same animal models, from mice studies, others.
  • Please include SD in the different figures (figures 2 to 6). In addition, a statistical analysis is required for figures 2 to 6 to support authors’ conclusions.
  • One of the biggest problems of multi serotype vaccines could be the induction of dominant responses against one antigen over the other, dampening the protection efficacy of the vaccine. In figure 7, data could suggest that antibody responses against serotype 1 is predominant or stronger than the antibody responses for other serotypes. Authors should include a better discussion/explanation for these data.
  • Although in figure 7 authors have examined the presence of nAb against the different serotypes, they should also analyzed (eg by ELISA) total IgG against the different serotypes (or VP2 proteins) or the presence of antibodies recognizing conserved epitopes of the VP2 proteins.
  • Have the authors evaluated the importance of T-cell responses in cross-protection using their pentavalent vaccine? Please, at least, comment this possibility in the discussion.

Author Response

1: The difference of used vaccine dose in sheep and cattle

Previously, 2 ml of 10*5 TCID50/ml monoserotype DISA8 has induced complete serotype specific protection in sheep (Feenstra et al., 2014ab).

We have observed a very similar seroconversion for VP7 Abs after vaccination with 1/10 of the standard dose and slightly lower response after 1/100 of the standard dose (van Rijn et al., 2017).

The same paper showed that the IM route induces a higher and lasting seroconversion compared to SC vaccination.

The IM route was successfully used in cattle (van Rijn et al. 2021).

Altogether, we decided IM vaccination of sheep with a lower vaccine dose.

Line 141: To clarify this approach/decision, we added the sentence: “Based on the similar seroconversion for VP7 Abs after immunization with 10-fold or 100 fold lower vaccine, we expected protection with a five-fold lower dose in pentavalent DISA12348 [58].”

However, the sheep trial showed incomplete protection, and therefore the standard dose was used for the cattle trial.

L144: We added after (For cattle, the vaccine dose was intended to be five times higher consisting 2 x 1ml of 5 x 105 TCID50/ml DISA12348 per vaccination): “, which corresponded to the standard dose for each serotype”.

L439: We stated: “In sheep, one fifth of the standard vaccine dose per serotype was studied to get in-sight with regard to the protective vaccine dose.”

2: SDs and statistical analysis

Although the SD can be calculated, a group of four animals cannot lead to a statistically significant result per assay or readout.

In order to minimize the number of animals according to the 3 Rs, we determined several parameters, such as fever, clinical score, viremia, and seroconversion for VP7 - and NS3 Abs and nAbs. All these were monitored during the entire trial, and more frequently after vaccinations and BTV challenge. The setup was similar as previous animal trials in sheep and cattle. In these trials, ‘multi-parameter monitoring’ of groups of four animals has resulted in conclusive results, since the different assays resulted in confirmatory results. For the animal trials described in this manuscript we successfully followed the same  vaccination-challenge and monitored the same parameters.

So, after discussions with the ethical committee, we agreed to use the minimum of four animals per group taking into account the measurement of multiple readouts and losing statistical analysis per assay/parameter. Indeed, we observed that results of different assays and observations are confirmatory similar as in previous animal trials. Furthermore, incomplete protection of sheep 2666 (vaccinated and challenged with BTV2 ) showing fever and loss of appetite was confirmed by viremia as measured with the Seg-1 and Seg-10 PCR test.

3: multi-serotype vaccines

Indeed, It has been shown that multi-serotype vaccines could affect the serotype specific immune response in a negative or positive manner. The latter, cross neutralization, is shown in Figure 1. Further, subsequent vaccinations with 3 different pentavalent LAVs (bottle A, B and C produced by OBP) showed nAb titers against serotypes not including in any of these bottles, and has been assumed to be protective against all 24 serotypes.

Here, we used the same DISA vaccine backbone with exchanged outer shell proteins. Still, the outer shell proteins can play a role in virus replication, as shown for serotype 3. Therefore, replication of DISA vaccine could also affect the immune response. Even more, different target species could react differently after DISA vaccination, as shown by the nAb titer for serotypes 4 and 8 in sheep and cattle.

More fundamental research is needed to unravel the contribution of each of the DISA vaccines as used in this pentavalent DISA vaccine. Still, we conclude that pentavalent vaccine protects against the studied serotypes 2 and 8, and likely protects against serotypes 1 and 4. The latter was based on the serotype specific nAb titers on the day of challenge. Animals were likely not protected against serotype 3, which is most likely caused by the instability and lower dose of DISA3 as discussed in the manuscript.

4: Yes, we strongly agree with this reviewer. It would be fantastic to have VP2 directed ELISAs available to investigate the Ab response in more detail. The same is true for IgG Abs directed against VP2. Indeed, this research will help to find conserved and serotype specific epitopes of nAbs. Unfortunately, these assays are not available yet, but progress has been made recently by production of different VP2proteins in plants (Fay et al., Viruses 2021).  So far, serotype specific nAb titers are the best parameters to predict protection against a certain BTV serotype, irrespective whether these are induced by the respective serotype or another related serotype.

5: Indeed, the importance of “other parts of the immune response“ have been discussed in previous papers, as referred in this manuscript. DISA vaccines are locally replicating and will induce full blown immune responses (innate immunity, B- and T-cell responses). Furthermore, immune responses will be directed against all BTV proteins, including more conserved viral proteins, except for NS3/NS3a protein. These immune responses will also contribute to protection.

L444: we stated “Possibly, the cell-mediated immunity plays an important role in protection after DISA vaccination."

Further, we checked the manuscript to improve the language and remove type errors as also requested by the other reviewers.

Reviewer 3 Report

The authors aimed to explore if Pentavalent Disabled Infectious Single Animal (DISA)/DIVA vaccine provides protection in sheep and cattle against different serotypes of bluetongue (BT) virus. Moreover, they studied both in sheep and cattle the application for different serotypes, in particular, pentavalent vaccine DISA12348, consisting of ‘European’ serotypes 1, 2 , 3, 4 and 8.

This a very interesting study with an adequate experimental methodology to support the data.

Abstract and Introduction section: better describe the aim of the study.

Conclusion Section: This paragraph required a general revision to eliminate redundant sentences and to add some "take-home" message.

Author Response

Abstract and introduction

To our opinion, the final sentence of the abstract harbors the aim as well as the “take-home” message: “The DISA/DIVA vaccine platform is flexible in use and generate monovalent as well as multivalent DISA vaccines to combat specific field situations with respect to Bluetongue.”

We added in the final paragraph of the Introduction section: Here, the DISA/DIVA vaccine platform was applied for different serotypes “aiming protection against multiple serotypes and eventually broad protection. As an example”, pentavalent vaccine DISA12348, consisting of ‘European’ serotypes 1, 2 , 3, 4 and 8, was studied in sheep and cattle.   

Conclusions

We improved the discussion section.

We also revised the last paragraph “Conclusions” and now contains several take-home messages:

“Pentavalent DISA12348 vaccine showed sterile immunity in sheep and cattle for studied serotypes 2 and 8 by lack of viremia and seroconversion against the DIVA target.”

“Detailed knowledge of core-out shell interactions will be required in order to develop DISA vaccines for eventually all BTV serotypes.”

“DISA vaccines will be suitable to combat specific emerging serotypes, to formulate tailor-made multivalent DISA vaccines, and eventually to formulate one or more multivalent DISA vaccines to achieve protection against all BTV serotypes.”

Further, we checked the manuscript to improve the language and remove type errors as also requested by the other reviewers.

Round 2

Reviewer 2 Report

Authors have addressed all the comments from the reviewers and the manuscript can be published in Vaccines.